# Sterile protection against *P. vivax* malaria by repeated blood stage infection in the *Aotus* monkey model

Nicanor Obaldía III[1,2,3] , Joao Luiz Da Silva Filho[3,4], Marlon Núñez[1], Katherine A Glass[5], Tate Oulton[5], Fiona Achcar[3,4], Grennady Wirjanata[6], Manoj Duraisingh[2], Philip Felgner[7], Kevin KA Tetteh[5], Zbynek Bozdech[6], Thomas D Otto[3] , Matthias Marti[2,3,4]

The malaria parasite *Plasmodium vivax* remains a major global public health challenge, and no vaccine is approved for use in humans. Here, we assessed whether *P. vivax* strain-transcendent immunity can be achieved by repeated infection in *Aotus* monkeys. Sterile immunity was achieved after two homologous infections, whereas subsequent heterologous challenge provided only partial protection. IgG levels based on *P. vivax* lysate ELISA and protein microarray increased with repeated infections and correlated with the level of homologous protection. Parasite transcriptional profiles provided no evidence of major antigenic switching upon homologous or heterologous challenge. However, we observed significant sequence diversity and transcriptional differences in the *P. vivax* core gene repertoire between the two strains used in the study, suggesting that partial protection upon heterologous challenge is due to molecular differences between strains rather than immune evasion by antigenic switching. Our study demonstrates that sterile immunity against *P. vivax* can be achieved by repeated homologous blood stage infection in *Aotus* monkeys, thus providing a benchmark to test the efficacy of candidate blood stage *P. vivax* malaria vaccines.

## Introduction

Malaria is caused by parasites of the genus *Plasmodium* that are transmitted to humans by the bite of the female Anopheles mosquito. Currently, ~241 million cases and 0.6 million deaths from malaria occur worldwide, an increase of 12% from the previous year (1). Most deaths are due to infection with *Plasmodium falciparum*, the most pathogenic of the species, especially in children under the age of five living in sub-Saharan Africa (1, 2).

After the elimination of *P. falciparum*, *Plasmodium vivax* is expected to remain a major cause of morbidity and mortality outside of Africa, especially in Central and South America, Asia, and the Pacific Islands (3, 4, 5). This is due in part to its peculiar biology, including silent parasite liver forms known as hypnozoites that can cause relapses and major parasite reservoirs in the bone marrow and spleen that may act as an unobserved pathogenic biomass and source for recrudescence (6, 7, 8, 9, 10, 11, 12). Complete removal of the parasite from the human reservoir is therefore challenging (4, 13), underscoring the need for innovative therapeutic strategies including the development of an effective vaccine (14, 15).

*P. vivax* malaria impacts the health of individuals of all ages causing repeated febrile episodes and severe anemia (14, 16), clinical severity including hemolytic coagulation disorders, jaundice, coma, acute renal failure, rhabdomyolysis, porphyria, splenic rupture (3, 17), and acute respiratory distress syndrome (18, 19, 20, 21). Fatal *P. vivax* cases are reported from all endemic regions across the globe (1, 14, 22). Compounding the epidemiology of the disease, *P vivax* malaria transmission is intermittent and acquired immunity is short and strain-specific (14). Even in low transmission regions, it is common to find individuals with asymptomatic parasitemia suggestive of natural premunition—a phenomenon resulting from a delicate host–parasite equilibrium in individuals with acquired immunity (14, 23). Epidemiological studies have demonstrated that repeated exposure increases clinical immunity and decreases parasite density and frequency of clinical episodes (24). For instance, individuals subjected to malariotherapy with *P. vivax* for treatment of neurosyphilis rapidly developed immunity after repeated blood stage infections (7, 25, 26, 27), and such repeated infection provided strain-transcending protection (24). Moreover, acquired immunity by repeated blood stage infection during malariotherapy has been reported in humans against *P. vivax*, *P. falciparum*, *Plasmodium ovale*, and *Plasmodium malariae* (27, 28, 29), providing an early benchmark for the feasibility of

[1]Departamento de Investigaciones en Parasitología, Instituto Conmemorativo Gorgas de Estudios de la Salud, Panamá City, Republic of Panamá [2]Department of Immunology and Infectious Diseases, Harvard TH Chan School of Public Health, Harvard University, Boston, MA, USA [3]Wellcome Centre for Integrative Parasitology, School of Infection and Immunity, College of Medical, Veterinary and Life Sciences, University of Glasgow, Glasgow, UK [4]Institute of Parasitology, Vetsuisse and Medical Faculty, University of Zurich, Zurich, Switzerland [5]Department of Immunology and Infection, London School of Hygiene and Tropical Medicine, London, UK [6]School of Biological Sciences, Nanyang Technological University, Singapore, Singapore [7]Institute for Immunology, University of California, Irvine, CA, USA

Correspondence: nobaldia@gorgas.gob.pa; Matthias.Marti@glasgow.ac.uk

developing a vaccine against *P. vivax* (7). However, understanding the correlates of protective immunity against *P. vivax* infection has proven difficult, mainly because of the lack of a continuous in vitro culture system for this parasite (30, 31).

The development of a vaccine against malaria with at least 75% protective efficacy is one of the two main objectives identified in the roadmap adopted by the global vaccine action plan until 2030 (32). Such an effective *P. vivax* vaccine should provide long-term and strain-transcending immunity. Current *P. vivax* vaccine studies are focused at inducing a stronger antibody response in combination with an already robust T-cell response (33, 34), based upon passive antibody transfer studies done in humans and laboratory animals (24, 35, 36). Nonetheless, to date, there is no vaccine against *P. vivax* approved for use in humans (14).

Several studies suggest that immunity to repeated blood stage infection in non-human primates is strain- and species-specific. For instance, Rhesus macaques immune to one strain of *Plasmodium knowlesi* may be partially susceptible to infection by another strain (35). Similar observations have been reported for *Aotus* repeatedly infected with *P. falciparum* blood stage parasites (37, 38). Interestingly, the same approach has produced heterologous cross-protection against *Plasmodium chabaudi* infection in mice (39). To assess whether strain-transcendent immunity can be achieved by repeated blood stage infection in *P. vivax*, we used the *Aotus* non-human primate model. The aims of our study were to determine (i) how many repeated homologous infections are required for control of parasitemia and the development of sterile immunity, and (ii) whether strain-transcending immunity could be achieved.

# Results

## *P. vivax* blood stage infection induces sterile immunity to homologous challenge

To evaluate the level of protection against repeated *P. vivax* blood stage infection, six *Aotus* monkeys (MN30014, MN30034, MN32028, MN32047, MN25029, and MN29012) were infected intravenously with 50,000 parasites of the *P. vivax* SAL-1 strain and monitored until peak parasitemia (Figs 1A and S1). The SAL-1 strain was originally isolated from a patient in El Salvador in the late 1960s and adapted to *Aotus* monkeys by W.E. Collins (40). During the first infection, all six animals were positive by day 6 post-inoculation (PI) and parasitemia increased steadily to more than $100 \times 10^3/\mu l$ infected red blood cells/$\mu l$ (mean ± SD = 100,198 ± 43,661/$\mu l$) until days 13–14 PI, when the animals were treated with a curative course of chloroquine (CQ) (Fig 1B). 65 d PI, one animal (MN29012) was removed from the study because of malaria-unrelated causes (Figs 1A and S1).

85 d PI, the remaining five animals (MN30014, MN30034, MN32028, MN32047, and MN25029) and the donor from the first inoculation (MN29041) were infected with SAL-1 using the same inoculum size of 50,000 parasites i.v. This time, by day 91 (day 6 PI of second inoculation, D6 PI II), all animals were positive by a blood smear, but parasitemia remained low with a mean peak of 2,332/$\mu l$ between days 94 and 95 (D9-10 PI II) (Fig 1B). A similar pattern was observed when total parasite load was measured by qRT-PCR (*18S rRNA*

levels) and parasite biomass by ELISA (pLDH levels) after this second inoculation (Fig S2). Two animals (MN32047 and MN30034) self-cured on day 98 (D13 PI II) and day 102 (D17 PI II), respectively, and a third animal (MN30014) became negative for 2 d between days 98 and 99 (D13-14 PI II) but recrudesced on day 100 (D15 PI II) and was treated with CQ on day 105 (D20 PI II) while still positive at the level of <10 parasites/$\mu l$. Meanwhile, MN29041 that had controlled its parasitemia until day 98 (D13 PI II) became negative on day 99 (D14 PI II), but recrudesced the next day, reaching a parasitemia level of 11,500/$\mu l$ on day 105 (D20 PI II) when it was treated with CQ (41). All animals received CQ treatment on day 105 (D20 PI II). Two animals (MN30034 and MN29041) were excluded after CQ treatment—MN30034 on day 169 (D114 PI II) because of severe anemia (Hct% = 20) and kidney failure, and MN29041 for malaria-unrelated causes on day 143 (D58 PI II). On day 166, the remaining 4 original animals (MN30014, MN32028, MN32047, and MN25029), plus a malaria naïve infection control (MN32029), were re-inoculated a third time with SAL-1 and followed up as described above (Fig 1B). This time, all animals except for the control (MN32029) that had a peak parasitemia of 95,550/$\mu l$ on day 179 (D13 PI III) remained negative and did not require CQ treatment. Of note, MN32028 had to be removed from the experiment on day 254 (D88 PI III) because of anemia and kidney failure. At necropsy, the animal presented with generalized subcutaneous edema (anasarca), with pericardial and pleural effusion, pulmonary edema, and evidence of chronic renal lesions. The cause of death was determined as renal failure (Fig 1A).

Altogether, these experiments demonstrate that repeated homologous *P. vivax* infection confers full protection (or sterile immunity) against a homologous challenge.

## Partial protection to heterologous challenge after repeated homologous infection

To determine the difference in protection between homologous and heterologous infections, we challenged on experimental day 276 the three remaining monkeys that went through three SAL-1 inoculations (MN30014, MN32047, and MN25029) plus a new malaria naïve infection control (MN31029) and the donor of the second SAL-1 infection (MN27050) with the CQ-resistant AMRU-1 strain (Fig 1A and B). The AMRU-I strain was originally isolated from a patient in Papua New Guinea in 1989 (42).

This time, all animals became positive. First, the two controls (MN27050 and MN31029) were positive on day 283 (D7 PI IV) with peak parasitemia of $131.5 \times 10^3/\mu l$ and $180 \times 10^3/\mu l$ on day 290 (D14 PI IV), respectively, when they were treated with MQ. Meanwhile, MN25029 became positive 4 d later on day 287 (D11 PI IV) with a lower (10-fold) peak parasitemia of $11.4 \times 10^3/\mu l$ on day 290 (D14 PI IV), clearing on day 296 (D20 PI IV) and treated with MQ on day 297 (D21 PI IV). Similarly, MN30014 became positive on day 295 (D19 PI IV) with a 100-fold lower peak parasitemia of 1,700/$\mu l$ on day 297 (D21 PI IV) compared with the peak parasitemia of the controls. The animal was treated with MQ on day 304 (D28 PI IV) for moderate anemia (Hct% = 27.4) and thrombocytopenia (PLT = $54 \times 10^3/\mu l$), while still positive at 1,510 parasites/$\mu l$. In contrast, MN32047 was positive only once on day 292 (D16 PI IV) with less than 10 parasites/$\mu l$ and was treated with MQ for severe anemia (Hct% = 16) on day 297 (D21 PI IV).

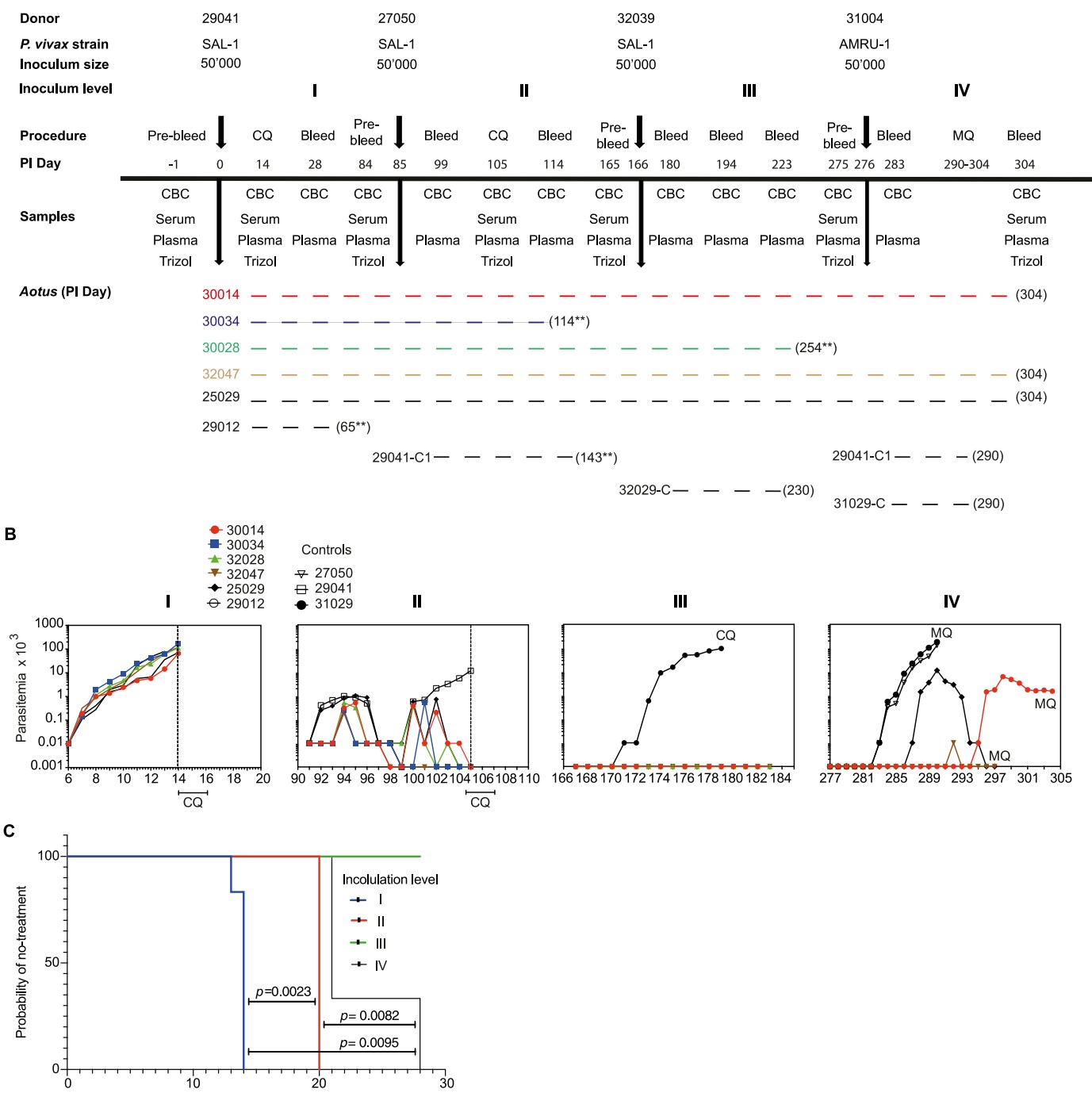

**Figure 1. Experimental timeline, parasite dynamics, and survival analysis.**
**(A)** Experimental timeline of infection and challenge. *: died of malaria-unrelated causes. **: anemia and renal failure. **(B)** Peripheral parasitemia across the experiment. Panels I–III show individual parasitemia of *Aotus* monkeys repeatedly infected with *P. vivax* SAL-1 (inoculations I–III). Panel IV shows *Aotus* challenged with *P. vivax* AMRU-1 (inoculation IV). Inoculated control animals were treated at peak parasitemia. **(C)** Probability of no treatment of *Aotus* repeatedly infected with the homologous *P. vivax* SAL-1 and heterologous *P. vivax* AMRU-1 strains at each inoculation level. *P*-values for survival curve comparison were obtained using the log-rank (Mantel–Cox) test. Survival curves for homologous infection 1 are shown in blue; homologous infection 2 in red; homologous infection 3 in green; and *P. vivax* AMRU-1 heterologous infection 4 in black. Complete blood count: red blood cell count. CQ, chloroquine, at 15 mg/kg oral for 3 d. MQ, mefloquine, at 25 mg/kg oral once. C, malaria naïve control. C1, control, once inoculated with *P. vivax*. PI, post-inoculation.

Altogether, this experiment revealed partial protection in 3/3 of the monkeys to a heterologous *P. vivax* challenge in sterile homologous immune animals. Partial protection was characterized by a delay of 4–12 d in patency and reduced parasitemia compared with the controls and a delay of 5–13 d in patency compared with the first homologous SAL-1 challenge. To further investigate the difference between repeated homologous and heterologous infections, we used survival analysis to assess the probability of the test subjects not requiring treatment at each inoculation level (Fig 1C). Median time to treatment was established at 14, 20, and none for homologous inoculation levels I–III, respectively, and 21 d for the heterologous challenge. Further analysis of various parasitemia-related parameters, including mean days patent, mean day of peak, mean peak parasitemia, and the total area under the parasitemia curve (AUC) (Fig S2), indicated that the level of protection against the heterologous challenge in inoculation level IV was similar to protection after one homologous challenge (i.e., inoculation level II). Indeed, the mean days of patency were shorter in infection level IV (unpaired *t* test = 3.060; df = 6; *P* = 0.0222), whereas the mean day to peak parasitemia was longer compared with level II (unpaired *t* test = 3.032; df = 6; *P* = 0.0230). No significant difference was found in peak parasitemia (unpaired *t* test = 2.191; df = 6; *P* = 0.0709) and AUC (unpaired *t* test = 2.409; df = 6; *P* = 0.0526) between levels II and IV (Fig S2).

### Severe anemia upon *P. vivax* heterologous challenge in sterile homologous immune *Aotus*

Next, we investigated the longitudinal dynamics of hematological parameters and selected blood chemistry during the repeated *P. vivax* infections (Fig 2 and Table S1). During the first inoculation, we observed a temporary but significant reduction in both hematocrit and platelet counts that coincided with peak parasitemia, as has been previously observed in *Aotus* (43) and humans experimentally infected with *P. vivax* (44) (Fig 2A–C, inoculation level I). During the second homologous infection, and with partial immunity ensuing, all the animals had hematological values within the normal range at peak parasitemia on day 20 PI, when they were treated with CQ for 3 d (Fig 2A–C, inoculation level II). Of note, MN25029 developed mild anemia (Hct% = 34.7) and severe thrombocytopenia (39 × 10$^3$/$\mu$l). During the third homologous infection, none of the animals became parasitemic and their hematocrit and platelet counts remained stable (note MN25029 again developed moderate thrombocytopenia [90 × 10$^3$/$\mu$l]) on day 14 PI (Fig 2A–C, inoculation level III). In contrast, the heterologous *P. vivax* AMRU-1 strain challenge triggered anemia and thrombocytopenia in all the animals (Fig 2A–C, inoculation level IV). For instance, mild-to-moderate anemia developed in two animals (MN30014 and MN32047) by day 7 PI, even though both animals had undetectable or subpatent parasitemia. Later, on day 28 PI MN30014 developed moderate anemia and severe thrombocytopenia with a parasitemia of 1,510/$\mu$l and needed treatment with MQ to end the experiment. Similarly, MN32047 developed severe anemia (Hct% = 19.3) on day 18 PI while still negative by light microscopy and needed treatment with MQ on day 21 PI to end the experiment. In contrast, MN25029 developed severe thrombocytopenia (24 × 10$^3$/$\mu$l) at peak parasitemia (11,430/$\mu$l) on

day 14 PI, even though its Hct% remained within normal limits (Hct% = 45), but later developed moderate anemia (Hct% = 26.2) on day 18 PI when it was still positive at <10 $\mu$l and was treated with MQ on day 21 to end the experiment.

Taken together, these data support previous studies observing the development of severe anemia (hematocrit < 50% of baseline) and thrombocytopenia (<50 × 10$^3$ × $\mu$l) in *P. vivax*-infected *Aotus* monkeys around days 12–15 PI (43). Indeed, 2/3 of the remaining original animals (MN30014 and MN32047) and a control (inoculated once) (MN27050) showed a Reticulocyte Production Index below 1.0 suggestive of bone marrow dyserythropoiesis (45) before inoculation level IV (Fig 2D), whereas only 1/3 of the original animals (MN25029) was over an Reticulocyte Production Index of 1.0 with a Hct% of 45.

### Antibody levels increase with repeated infections

In the next series of experiments, we analyzed the development of antibodies against a crude *P. vivax* lysate across repeated infections (Fig 3A, Table S2). After the first inoculation with *P. vivax*, SAL-1 total antibody (Ab) levels reached a mean of 3.1 log$_{10}$ arbitrary ELISA units (day 28 PI), decreasing slightly to 2.9 log$_{10}$ ELISA units by day 84 PI. After the second homologous inoculation (day 84 PI), Ab levels peaked at 4 log$_{10}$ ELISA units on day 114 PI, decreasing slightly again to 3.5 log$_{10}$ ELISA units by day 165 PI (Fig 3B). After the third homologous inoculation (day 165 PI) when all the animals were sterile-protected against challenge (Fig 3B), Ab levels remained over 4.0 log$_{10}$ ELISA units until day 275 PI when the animals were challenged with the heterologous *P. vivax* AMRU-1 strain. This time, a booster response was observed with Ab levels increasing to 4.3 log$_{10}$ ELISA units by day 304 PI (Fig 3B). Interestingly, Ab levels appear to be negatively correlated with parasitemia (Fig 3C). In summary, the dynamics of mean parasitemia and ELISA titers during inoculation levels I–IV suggest that an ELISA titer of 3–4 arbitrary log$_{10}$ units would fully protect against challenge with a homologous but only partially protect against a heterologous strain of *P. vivax* (Fig 3D). These correlates of protection provide a benchmark for efficacy testing of *P. vivax* blood stage candidate vaccines in the *Aotus* model.

### Quantification of antigen responses using a *P. vivax* protein microarray

Antibody responses during repeated infection (log$_2$ [antigen reactivity/no DNA control reactivity]) show that 66 of 244 *P. vivax* antigens in the protein microarray demonstrated reactivity above 0 in 10% of all samples analyzed (Fig 4A). When we compared the antibody levels for these 66 antigens for all time points in inoculation III (the final homologous challenge) versus inoculation IV (the heterologous challenge) for the three monkeys that completed the entire experiment, there were no differentially reactive antigens (paired *t* test with an FDR correction). It is possible that there are differentially reactive antigens that were not identified in this study because of the limited number of antigens tested and/or the small sample size.

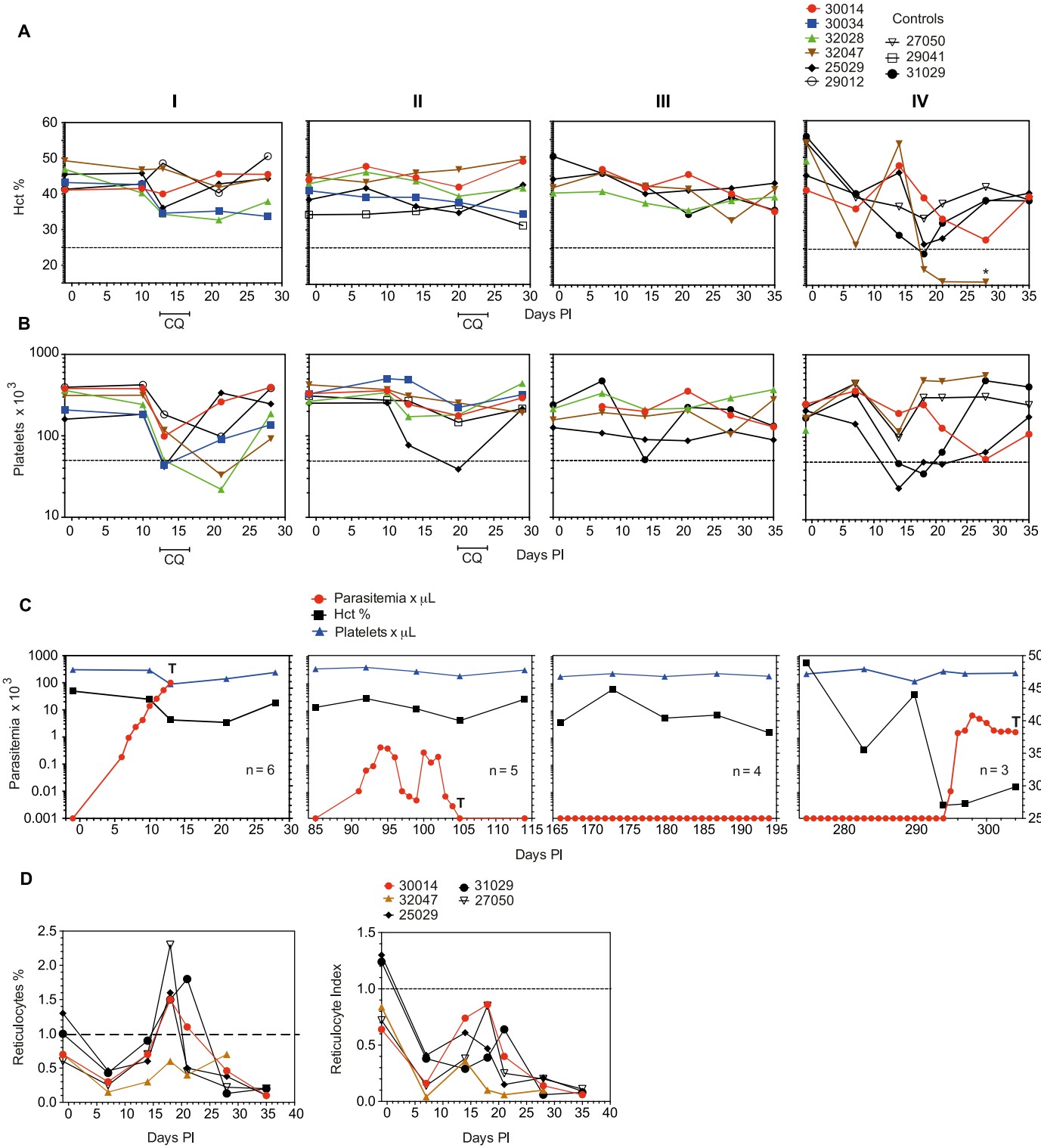

**Figure 2. Hematological and parasite parameters.**
Panels (A, B, C) show hematocrit levels (Hct%) (A), platelet counts (B), and combined data from (A, B) and mean parasitemia (C) across inoculation levels I–IV. Panel (D) shows the percentage of reticulocytes and the Reticulocyte Production Index at infection level IV. RPI = Reticulocyte Absolute Count/Reticulocyte Maturation Correction. **(C)** Reticulocyte Absolute Count = Hct%/45 x reticulocyte %. T = CQ: chloroquine, at 15 mg/kg oral for 3 d; and MQ at 25 mg/kg once for rescue treatment of *P. vivax* AMRU-1 infections in panel (C).

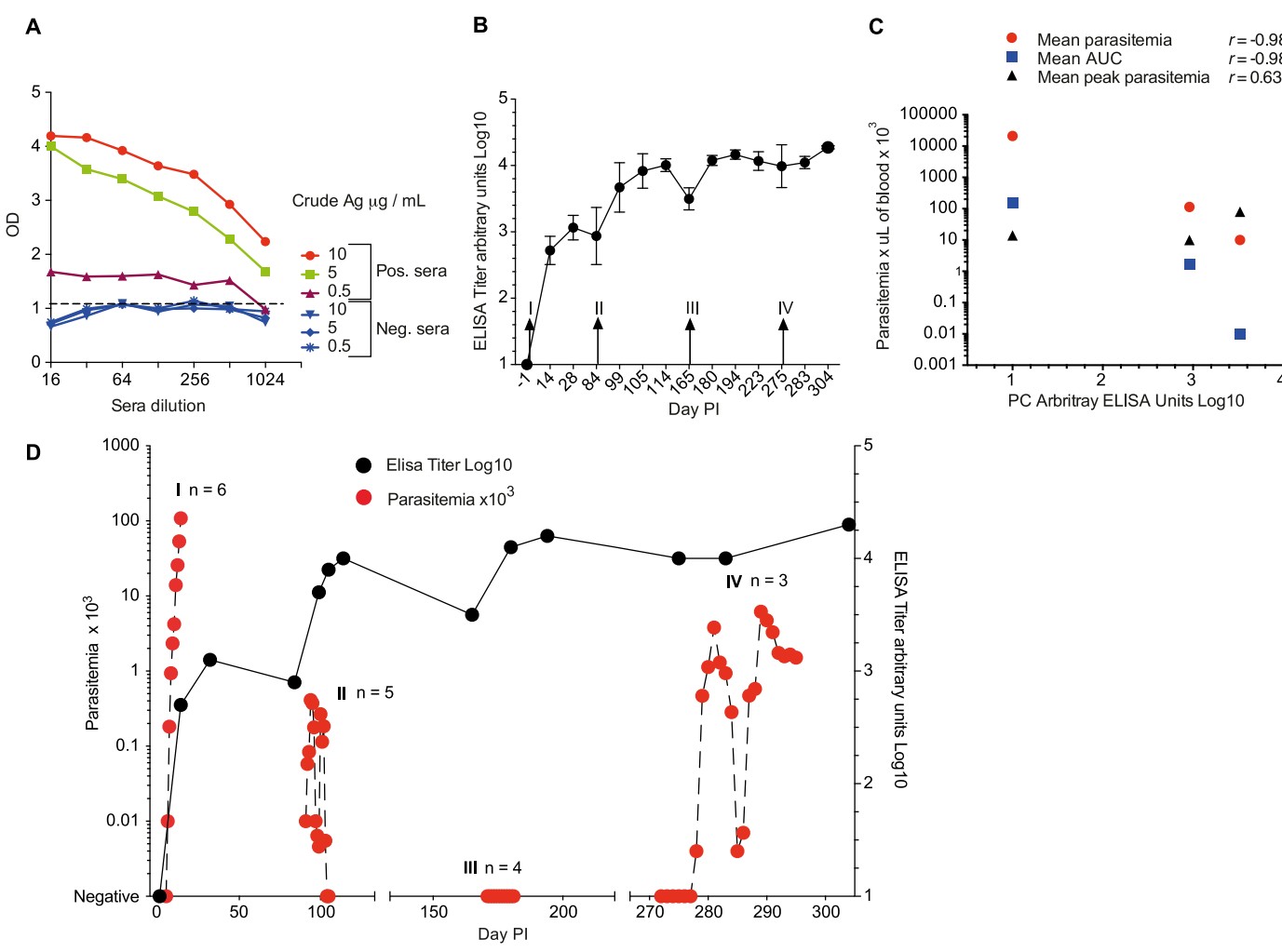

**Figure 3. ELISA titers of *Aotus* repeatedly infected with *P. vivax* blood stages.**
**(A)** Crude antigen checkerboard titration. *P. vivax* SAL-1 antigen was prepared from *Aotus*-infected red blood cells purified by Percoll cushion (47%) centrifugation and adsorbed to the plate wells diluted in PBS, pH 7.4, at a concentration of 5 μg/ml. Secondary antibodies (peroxidase-conjugated goat anti-monkey, Rhesus macaque) were diluted 1:2,000 in PBS, pH 7.4, and optical density (OD) was read using a 492-nm filter. **(B)** Mean ELISA* titers of *Aotus* immunized by repeated infection with the homologous SAL-1 and challenged with the heterologous AMRU-1 strains of *P. vivax*. I–IV indicate inoculation levels, each with an inoculum of 50 × 10³ infected red blood cells. Levels I-III indicate infection with homologous SAL-1. Level IV indicates infection with heterologous AMRU-1. **(C)** Pearson's correlation analysis of mean ELISA titers at inoculation levels I (n = 6), II (n = 5), and III (n = 4) showed a high negative correlation versus mean parasitemia ($r = -0.98$), the mean area under the curve ($r = -0.98$), and a moderate positive correlation versus mean peak parasitemia ($r = 0.63$). **(D)** Combined plot of mean parasitemia and ELISA titers with *Aotus* repeatedly infected with the homologous SAL-1 (infection I–III) and challenged with the heterologous AMRU-1 (infection IV).

Within inoculation levels I–III, the number of reactive antigens (antigen breadth) was significantly increased at days 14, 21, and/or 28 when compared to the pre-inoculation antigen breadth (Fig 4B, $P < 0.05$, Wilcoxon's matched-pairs test). The trend for increased antigen breadth over time is similar but non-significant for the heterologous infection with the *P. vivax* AMRU-1 in inoculation IV. When we calculated the area under the curve of antigen breadth for each inoculation level for the three monkeys, which completed all four inoculations, inoculation III and IV were both significantly higher than inoculation I (and were not different from each other) (Fig 4C, $P < 0.05$, repeated-measures ANOVA with paired-sample post hoc $t$ tests). These data show that repeated infections of the homologous strain *P. vivax* SAL-1 (inoculation levels I–III) increase the breadth of the immune response as the number of infections increased, and that

the breadth remained (but did not increase further) high during heterologous challenge with *P. vivax* AMRU-1. Those antigens eliciting the strongest immune response also showed the strongest positive correlation with ELISA titers (Fig S3). Interestingly, no negative association with parasite parameters was observed, whereas similar sets of antigens showed significant negative correlations with platelet counts (Fig S3). These include two MSP1 peptides (PVX_099980), an early transcribed membrane protein (ETRAMP) peptide (PVX_090230), and peptides to two exported proteins (PVX_121935 and PVX_083560). We also found that the ELISA titer for the crude lysate correlated well with antigen breadth; however, correlations were only significant at inoculation level II on days 99 (Pearson's R = 0.98, significant at $P < 0.005$) and 114 (Pearson's R = 0.86, trend at $P = 0.062$) (Figs 4D and S4).

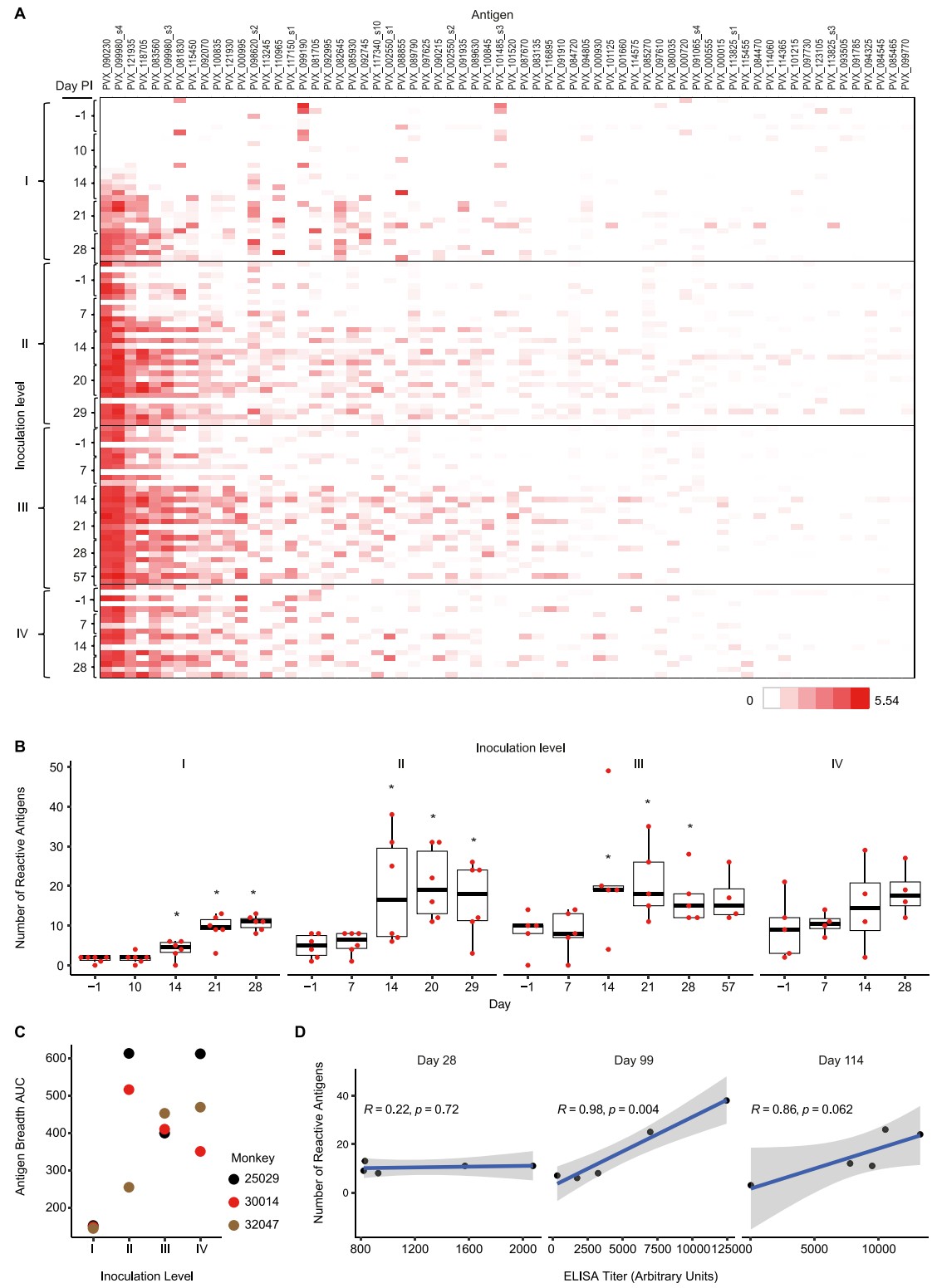

**Figure 4. Protein microarray.**
**(A)** Shown are antibody responses ($\log_2$ [antigen reactivity/no DNA control reactivity]) to 66 of 244 *P. vivax* IVTT antigens with reactivity above 0 in 10% of all samples and across monkeys. Thus, zero represents equal or lower reactivity than the mean of the no DNA control spots. Antigens are ordered from the highest to the lowest overall mean. Samples are ordered top to bottom by inoculation level, day, and then monkey. **(B)** Antigen breadth (number of *P. vivax*-reactive antigens) by post-infection day at each inoculation level (I–IV). Antigens were considered reactive if the reactivity was higher than the mean + 3 SD of the no DNA control spots for that sample. * indicates a significantly higher antigen breadth at that day than at baseline (day -1) within each inoculation (*P* < 0.05, Wilcoxon's matched-pairs test, one-sided). **(C)** Area under the

A longitudinal follow-up during repeated infection revealed major immunogenic antigens (Ags) by protein microarray. Indeed, seven targets have significantly higher antibody responses at inoculation level III compared with inoculation level I (Fig S5, Table S3), including the ETRAMP (PVX_090230), parasitophorous vacuolar protein 1 (PV1) (PVX_092070), merozoite surface protein 1 (MSP-1) (PVX_099980, fragments 2 and 3), and three *Plasmodium* exported proteins (PVX_121930, PVX_083560, and PVX_121935). The maintenance of antigen breadth after heterologous challenge (inoculation IV) may suggest the presence of homologous or cross-reactive antigens between the two isolates. However, amino acid sequences for all seven targets (six genes) were identical, except for a region of 14 amino acids in one of the *Plasmodium* exported proteins (PVX_083560). Altogether, these data suggest that sterile protection upon homologous challenge and partial protection upon heterologous challenge may not be due to these proteins; however, they may be used as correlates of protection.

### Genetic diversity rather than immune evasion determines the level of strain-transcendent protection

Our data so far suggest that protection from the homologous challenge is antibody-mediated; however, the limited resolution of the ELISA and protein array data cannot explain the lower protection after the heterologous challenge. As an alternative approach, we investigated possible immune evasion mechanisms on the genomic and transcriptional level.

For this purpose, we performed whole-genome sequencing of both SAL-1 and AMRU-1 strains to improve the strain-transcendent coverage of the existing *P. vivax* microarray platform (46). Selective whole genome amplification enabled targeted amplification of the AT-rich subtelomeres of AMRU-1 and SAL-1 strains used in this study (Fig 5A). Assembly and annotation generated continuous subtelomere sequences for SAL-1 and AMRU-1. The number of contigs in the original SAL-1 dropped from 2,748 to 113, highlighting the continuity of the PacBio assembly (Fig 5B). After the annotation with Companion (47), the improved assembly increased the number of *pir* genes for SAL-1 from 124 to 425, demonstrating that long reads better represent the number of variable gene families in subtelomeric regions. A comparison of the *Plasmodium* interspersed repeat (*pir*) gene repertoire across strains revealed 593 and 425 *pir* genes in AMRU-1 and SAL-1, respectively, compared with over 1,000 in the PvP01 reference strain (Fig 5C). This difference in number may be because the reference strain came straight from patient infection, whereas SAL-1 and AMRU-1 may have adapted during repeated passages through monkeys. Finally, the proportion of *pir* subtypes remains constant across strains as previously reported (48).

With this information in hand, we complemented the existing microarray probe set that was generated for the *P. vivax* core genome (i.e., all the chromosome central regions (46)) with probes for the SAL-1 and AMRU-I subtelomeric genes. Next, we investigated whether the virulent phenotype upon heterologous AMRU-1 infection was a result of immune evasion. Differential gene expression (DGE) analysis and principal component analysis of the expressed genes revealed greater differences in both core and *pir* genes when comparing AMRU-1 parasites from heterologous challenges (after three inoculations with SAL-1) to SAL-1 parasites during the homologous challenges (Fig 6A—left panel, Fig 6B). We also compared the DGE of AMRU-1 parasites between animals previously infected with three SAL-1 inoculations with (i) animals previously infected with only one SAL-1 inoculation and (ii) the malaria naïve infection control. Interestingly, only a small number of changes in core and *pir* genes were observed across these comparisons (Fig 6A—right panel). The clear overall similarity of sample distribution in the principal component analysis plots based on the DGE of core (Fig 6B—left panel) or *pir* genes (Fig 6B—right panel) suggests that the repeated SAL-1 infections do not induce extensive *pir* gene switching either in SAL-1 or in AMRU-1 parasites. Rather, there appear to be significant strain-specific differences in both core and *pir* expression between SAL-1 and AMRU-1. Further analysis using a *pir* gene network revealed no apparent changes in *pir* gene expression across SAL-1 challenges or upon the AMRU-1 challenge (Fig 6C).

Altogether, the transcriptional analysis does not indicate that AMRU-1 parasites actively evade the antibody-mediated protection induced by SAL-1 homologous challenges by antigenic switching. Thus, the lower protection observed after the heterologous challenge may be due to major genetic differences and hence antibody epitope variation between these two geographically separated strains (49).

## Discussion

Previous trials of *P. falciparum* and *P. vivax* vaccine candidates have demonstrated the utility of the *Aotus* model in supporting vaccine development (50, 51, 52). Various asexual stage vaccine candidate antigens have been subjected to testing in *Aotus* (51, 53, 54, 55, 56, 57, 58, 59, 60, 61), but only a few have shown some level of efficacy in human clinical trials (14, 62). The development of highly effective strain-transcendent immunity against malaria is a universal goal of vaccine developers (63). Recently, whole-organism blood stage malaria vaccines have gained prominence as an alternative to subunit vaccines (64, 65). One major advantage of vaccination using whole blood stage parasites is the multiplicity of immunogenic antigens, including those conserved across strains that may be able to induce strain-transcendent immunity (66, 67).

To assess whether strain-transcendent immunity can be achieved by repeated blood stage infection with *P. vivax*, and to investigate possible correlates of protection during repeated infection, we infected six *Aotus* monkeys with the *P. vivax* SAL-1 strain until sterile-protected and then challenged with the AMRU-1. We demonstrate that repeated whole blood stage infection with a homologous *P. vivax* strain (i.e., same strain) induces sterile immunity in *Aotus* monkeys after only two infections. In contrast,

---

curve of the antigen breadth at each inoculation level for the three monkeys that completed the experiment. **(D)** Pearson's correlation of ELISA titer at each day post-infection versus antigen breadth. *P*-values shown are from *t* tests with the null hypothesis that the correlation coefficient equals 0.

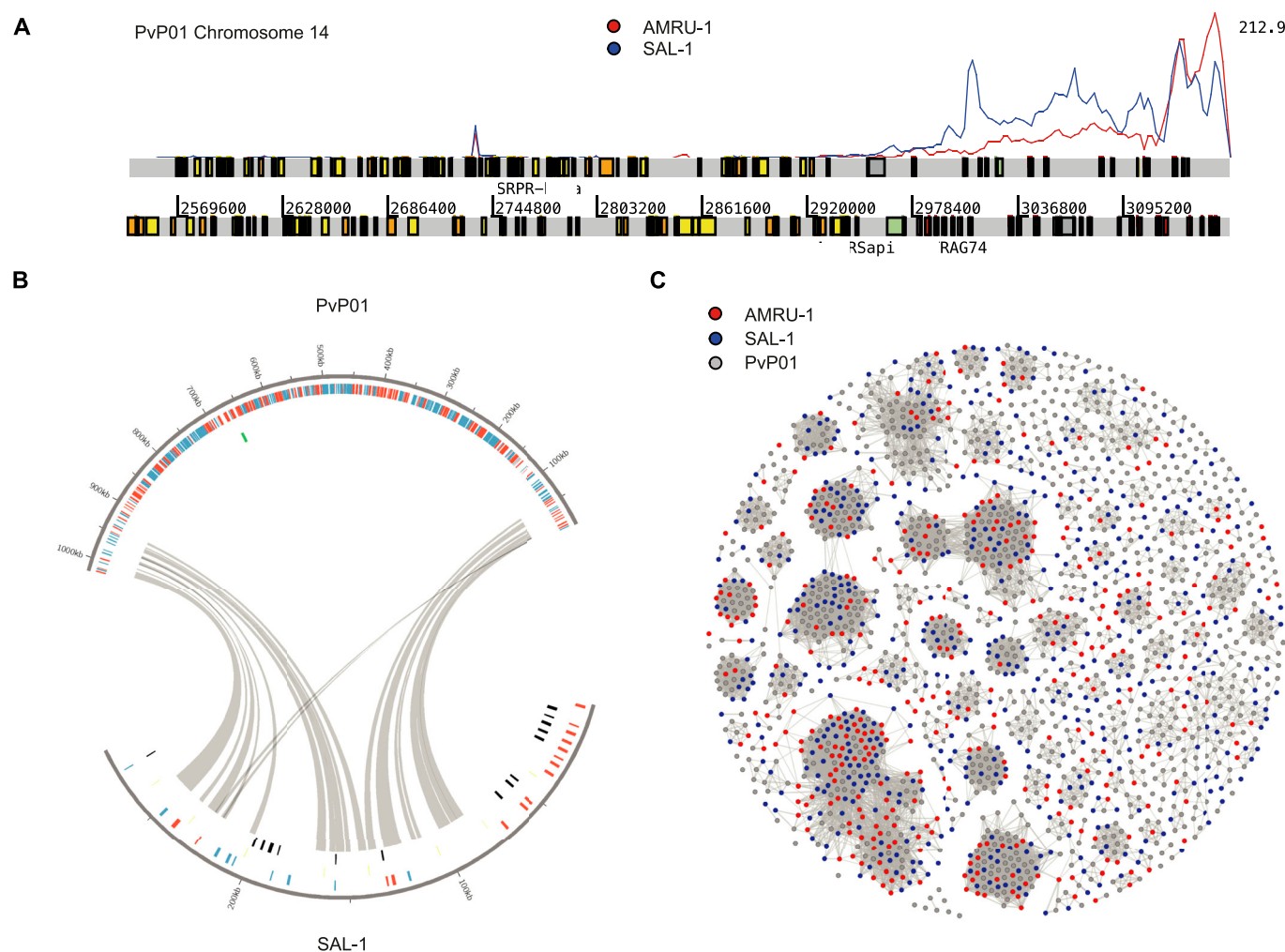

**Figure 5. Selective whole-genome amplification (sWGA) and PacBio sequencing of *Plasmodium vivax* SAL-1 and AMRU-1.**
**(A)** Artemis screenshot. Shown is one arm of *P. vivax* PvP01 chromosome 14, with PacBio reads mapped (SAL-1 in blue and AMRU-1 in red). Most of the coverage occurs in subtelomeric regions, demonstrating the specificity of the selective whole-genome amplification. **(B)** Circos plot of one representative SAL-1 contig that contains mostly *pir* genes. The contig maps to chromosome 1 of *P. vivax* reference PvP01. Gray lines show syntenic matches of *pir* genes between the two strains. **(C)** Gephi plot showing *pir* genes from AMRU-1 (red), SAL-1 (blue), and PvP01 reference (gray). Genes are connected if they share at least 32% global identity.

*Aotus* monkeys infected with *P. falciparum* needed between three and four (68) and between six and seven (37) repeated infections, respectively, to achieve sterile immunity. This is consistent with previous observations made during malariotherapy in patients with neurosyphilis, demonstrating that immunity to *P. falciparum* is acquired more slowly than to *P. vivax* or *P. malariae* (24). Interestingly, *Saimiri sciureus boliviensis* monkeys immunized with irradiated sporozoites of *P. vivax* SAL-1 and challenged four to nine times with homologous viable sporozoites over a period of almost 4 yr showed sterile protection (69). However, all animals remained susceptible when challenged with SAL-1 blood stage parasites, suggesting that humoral immunity is a correlate of protection against repeated blood stage infections.

Furthermore, our study demonstrates that the sterile immunity achieved after repeated infection with a homologous strain was only partially protective after a heterologous challenge (i.e., delay to infection and reduction in peak parasitemia compared with the control). Similar observations have been reported for *P. falciparum* in *Aotus* (37). In both cases, heterologous challenges resulted in severe anemia and thrombocytopenia irrespective of the parasitemia level. Such hematological manifestations in semi-immune *Aotus* monkeys with low or subpatent *P. falciparum* parasitemia have been attributed in the past to clearance of non-infected RBCs mediated by autoantibodies (70, 71, 72, 73), sequestration of infected RBCs, bone marrow suppression (70, 71), and immune-mediated thrombocytopenia (43, 74, 75, 76). The pernicious severe anemia without thrombocytopenia observed in the monkey MN32047 during subpatent parasitemia may have been the result of immune complex disease, or of dyserythropoiesis because of bone marrow infection, as previously described in humans and *Aotus* monkeys infected with *P. falciparum*, *P. malariae*, and *Plasmodium brasilianum* (74, 77, 78, 79, 80, 81, 82, 83).

A *P. vivax* in vitro transcription/translation (IVTT) reaction protein microarray revealed antibody responses against major

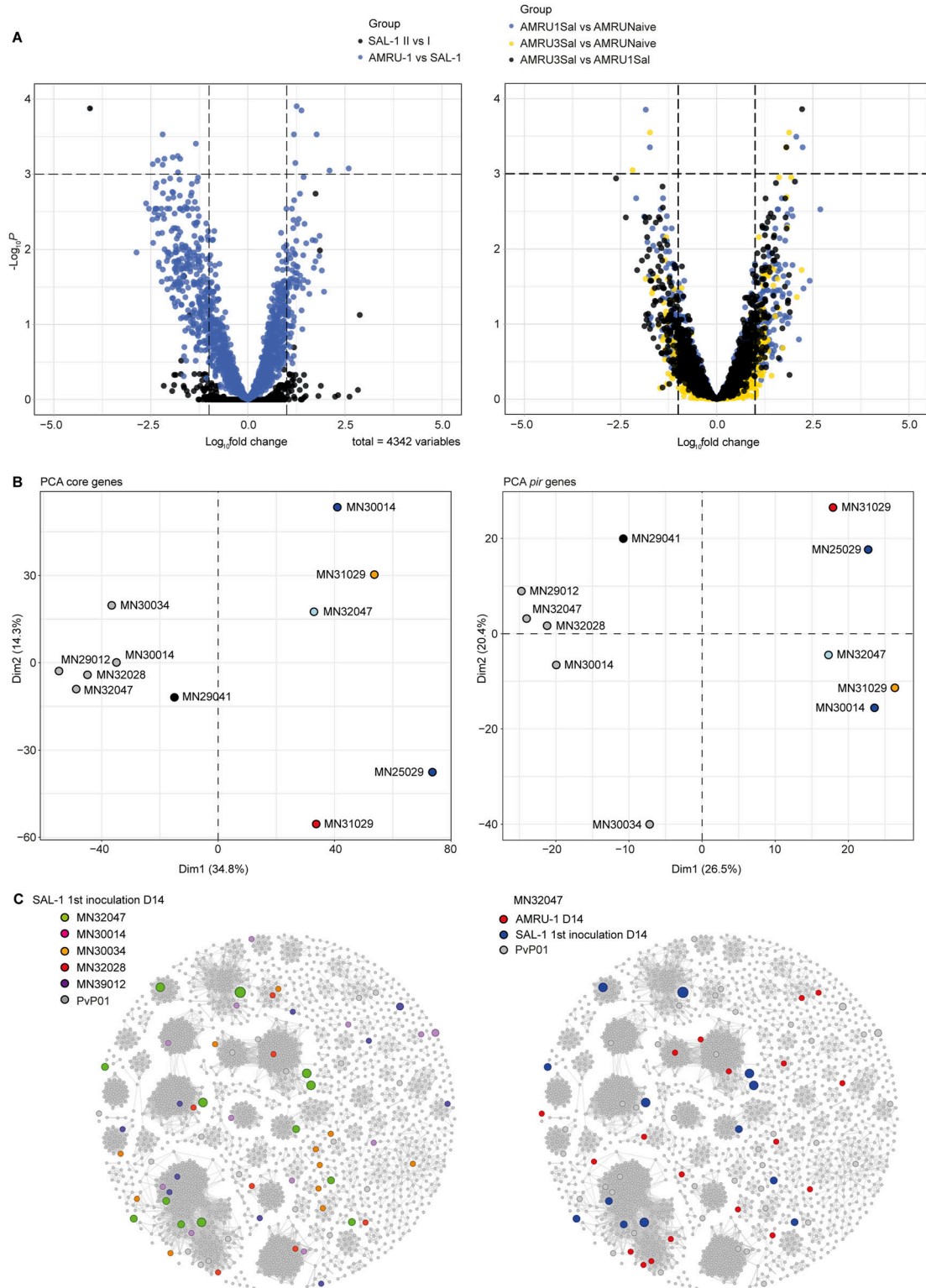

**Figure 6. Parasite gene expression comparisons across infection regimes.**
**(A)** Differential gene expression (DGE) across core genes. Volcano plots show DGE between infection regimes. *Left*: DGE of core genes between SAL-1 inoculation II versus SAL-1 inoculation I (black) and between AMRU-1 inoculation IV versus the averaged expression of SAL-1 during the homologous challenges (blue). *Right*: DGE of core genes across AMRU-I infection regimes. Yellow dots represent DGE between AMRU-1 parasites from *Aotus* monkeys previously infected with three SAL-1 inoculations (AMRU3Sal) versus AMRU-1 parasites from naïve *Aotus* monkeys (AMRUNaive). Black dots represent DGE between AMRU3Sal versus AMRU-1 parasites from *Aotus* monkeys previously infected with only one SAL-1 inoculation (AMRU1Sal). Blue dots represent DGE between AMRU1Sal versus AMRUNaive. Each dot represents one annotated *P. vivax* core gene and is displayed according to the fold change in expression (x-axis, in log₂) and statistical significance (y-axis, in negative logarithm to the base 10 of the

immunogenic blood stage antigens (Ags) in this study. This approach was supported by previous studies, which demonstrated the utility of the IVTT platform in high-throughput antigen discovery across several disease areas (84, 85, 86, 87, 88). Immune reactivity to individual antigens and antibody breadth in sera from these animals increased with each inoculation level and were statistically significantly different between inoculation levels I and III, when the animals achieved sterile immunity to a homologous SAL-1 challenge. Among the most significant asexual blood stages, antigens detected by the protein microarray were ETRAMP (PVX_090230) located in chromosome 5, and two MSP1 fragments: PVX_099980_s4 and PVX_099980_s2 located in chromosome 7, the latter is a leading vaccine candidate that has been identified as a major determinant of strain-specific protective immunity (89).

In this study, animals with sterile immunity to a *P. vivax* SAL-1 homologous challenge were partially protected against a heterologous AMRU-1 strain. This difference in protection may have been the result of cross-reactive but polymorphic antigens associated with essential parasite phenotypes such as red cell invasion, rosetting, or cytoadherence. Maintaining genetic diversity enables immune evasion, as suggested in recent genomic studies of *P. vivax* parasites from distinct geographic origins such as SAL-1 and AMRU-1 (49, 90). Finding conserved and cryptic (not exposed to the immune system) epitopes involved in essential phenotypes that could be targeted by strain-transcending neutralizing antibodies represents a possible way forward (91). In contrast to *P. falciparum* that uses the variant PfEMP1 antigens to induce cytoadherence and avoid splenic clearance of blood stage parasites, limited vascular sequestration occurs in most other *Plasmodium* species investigated so far. At least in *P. vivax*, this process may be mediated by *P. vivax* orthologs of the PIR variant antigens (48). In our study, gene expression analysis along multiple infections allowed correlating *pir* gene expression with the immune response across infections to illuminate parasite immune evasion mechanisms during the heterologous challenge. Interestingly, only minor changes in *pir* gene variant expression were observed across all the different inoculation levels, whether homologous or heterologous. Further analysis using a *pir* gene network confirmed no apparent changes in *pir* gene expression in AMRU-1 parasites, regardless of the nature of the previous infections. Together, the transcriptional analysis does not indicate that *P. vivax* actively evades the antibody-mediated protection through antigenic switching. These findings are in accordance with previous studies that have shown no significant difference in the antibody response against PIR antigens when comparing the sera of single versus repeated infection in patients for, and hence, they question the role of PIR antigens in *P. vivax* immune evasion (92, 93). The partial protection observed in the heterologous AMRU-1 challenges may therefore be due to major

genetic differences and hence antibody epitope variation between the two strains (49). In comparing the SAL-1 and AMRU-1 strains with the PvP01 reference strain, the sequence data demonstrated clear differences between the isolates in the whole-genome analysis. Therefore, this suggests that the current iteration of the microarray (n = 244) used in the study did not capture the sequence target(s) responsible for the partial protection observed. To overcome this limitation and induce high levels of protective antibodies, we propose the use of an immunization regime with whole parasite antigen pools from a mixture of genetically diverse strains. Another limitation of this study is the small number of subjects. The study can be considered as exploratory (i.e., looking for patterns of response rather than hypothesis testing (94)); hence, the number of subjects used in the only group studied is typical of such exploratory research with humans (34, 95) and NHPs (37).

In conclusion, our study demonstrates that sterile immunity against *P. vivax* can be achieved by repeated homologous blood stage infection in *Aotus* monkeys. It also contributes to our understanding of the pathogenesis of *P. vivax*-induced anemia, *P. vivax* asexual blood stage antigen discovery and correlates of protection, and possible immune evasion mechanisms. Most importantly, we establish a benchmark for *P. vivax* protective immunity in the *Aotus* monkey model, providing an important criterion for vaccine development (37).

# Materials and Methods

### Ethics statement

The experimental protocol entitled "Induction of sterile protection by blood stage repeated infections in *Aotus* monkeys against subsequent challenge with homologous and heterologous *P. vivax* strains" was approved and registered at the ICGES Institutional Animal Care and Use Committee (CIUCAL) under the accession number CIUCAL-01/2016. The experiment was conducted in accordance with the Animal Welfare Act and the Guide for the Care and Use of Laboratory Animals of the Institute of Laboratory Animal Resources, National Research Council (96), and the laws and regulations of the Republic of Panamá.

### Animals and parasites

Twelve laboratory-bred (lab-bred) adult male and female "spleen-intact" *Aotus l. lemurinus lemurinus* Panamanian owl monkeys of karyotypes VIII and IX were used in the study (97). The animals were cared for and maintained as described elsewhere (98). Isolates of *P. vivax* SAL-1 originally adapted to splenectomized *Aotus* monkeys by

---

*P*-value). **(B)** Principal component analysis of the parasite core gene (left panel) and *pir* gene (right panel) expression profiles from each biological replicate, colored according to the corresponding group: SAL-1 parasites at day 14 PI of the first inoculation (gray); SAL-1 parasites at day 14 PI of the second inoculation (black); AMRU-1 parasites at day 14 PI from *Aotus* monkeys previously infected with three SAL-1 inoculations (light blue dots); gene expression of AMRU-1 parasites at day 28 PI from *Aotus* monkeys previously infected with three SAL-1 inoculations (blue dots); gene expression of AMRU-1 parasites at day 1 PI from naïve *Aotus* monkeys (orange dots); and gene expression of AMRU-1 parasites at day 14 PI from naïve *Aotus* monkeys (red dots). **(C)** *pir* gene network analysis comparing *P. vivax pir* gene expression in SAL-1 versus AMRU-1 infections in *Aotus* monkeys. The same network as in Fig 5C, except that larger circles indicate the *pir* gene expression level. Left panel: *pir* expression in SAL-1 parasites at day 14 PI of the first inoculation across individual monkeys. Right panel: comparison of *pir* expression in monkey MN32047 between SAL-1 parasites at day 14 PI of the first inoculation and AMRU-1 parasites at day 14 PI of the fourth inoculation.

W.C. Collins in 1972 (40) and further adapted to spleen-intact *A. l. lemurinus* (43, 51), and of *P. vivax* AMRU-1 from Papua New Guinea originally adapted to splenectomized *Aotus* by R.D. Cooper in 1994 (99, 100) and further adapted to spleen-intact *A. l. lemurinus* by Obaldía N.III. in 1997 (101) were used.

Briefly, each frozen stabilate of SAL-1 and AMRU-1 was thawed, washed three times with incomplete RPMI medium, and resuspended in 1 ml of RPMI medium. This suspension was used for intravenous (i.v.) inoculation into the saphenous vein of a donor animal using a 25-gauge butterfly needle catheter attached to a 3-ml syringe. When the level of parasitemia reached a peak around days 12–15 post-inoculation (PI), a dilution of blood was made in RPMI to get a total inoculum of 50,000 parasites/ml. All animals received 1 ml of the inoculum through the saphenous vein.

In total, 12 spleen-intact lab-bred animals were used in this experiment. Six monkeys (three males and three females; MN30014, MN30034, MN32028, MN32047, MN25029, and MN29012) were repeatedly infected with *P. vivax* SAL-1 (homologous challenge) three times (levels I–III), and another six animals served as either donors or were assigned as infection or naïve controls. Donors and controls were reassigned back into subsequent inoculation levels as depicted in Figs 1A and S1. Three SAL-1 homologous sterile immune monkeys from the original six, plus one infected once with SAL-1 and one malaria naïve control, were rechallenged (level IV) with the heterologous CQ-resistant and *Aotus*-adapted *P. vivax* AMRU-1 strain (43). The animals were treated with CQ at 15 mg/kg for three consecutive days during inoculation levels I-III, and a drug wash-out period of 70 d was kept between inoculation levels I and II and 65 d between levels II and III. No CQ treatment was instituted in inoculation level III. To treat the *P. vivax* AMRU-1 CQ-resistant strain, inoculated animals on inoculation level IV and at the end of the experiment were treated with MQ at 25 mg/kg orally once.

### General procedures

5 d after infection, the animals were monitored for any signs of clinical disease and bled 5 $\mu$l from a prick made with a lancet in the marginal ear vein to measure daily parasitemia. Parasitemia was determined using a thick blood smear stained with Giemsa as described in the Earle and Perez (1932) technique (102). Blood samples were also collected at regular intervals from the femoral vein to assess humoral immune responses against *P. vivax* blood stage proteins, for complete blood count and blood chemistry (liver and renal panel), for collection of parasite DNA on FTA Elute cards (Whatman), and for RNA in TRIzol solution (Invitrogen) for molecular biology studies. The animals were treated with mefloquine (MQ) at 25 mg/kg orally by gastric intubation to end the experiment.

### Criteria for parasitemia

For this study, patency was defined as the first of three consecutive positive days after inoculation. Clearance was defined as the first of three consecutive negative days. Recrudescence was defined as the first of three consecutive positive days after a period of clearance. Positivity of <10/$\mu$l for less than 3 d was considered evidence of subpatent infection.

### Criteria for anemia and thrombocytopenia

For this study, we classified anemia based on the hematocrit % as mild (Hct% = 31–36), moderate (Hct% = 25–30), or severe (Hct% < 25). Thrombocytopenia was considered mild if platelet counts were between 149 and 100 × $10^3$/$\mu$l, moderate if they were between 99 and 50 × $10^3$/$\mu$l, or severe if they were <50 × $10^3$/$\mu$l.

### Drug treatment

CQ was administered orally for three consecutive days at 10 mg/Kg daily at peak parasitemia. Rescue treatment with MQ was triggered if the hematocrit reached 50% of baseline or hemoglobin was <8 gm/dl, platelets were <50 × $10^3$/$\mu$l, or the animals remained positive by LM after day 28 PI (43).

### Serology

#### *Serum ELISA*

The *P. vivax* SAL-1 antigen was prepared from *Aotus*-infected red blood cells purified by Percoll (GE Healthcare Bio-Sciences AB) cushion (47%) centrifugation as described (103), and adsorbed at 5 $\mu$g/ml concentration diluted in PBS, pH 7.4, to a 96-well plate at 4–8°C overnight. The plates were blocked with 5% skimmed milk in PBS/0.05% Tween for 2 h. Serum samples were added to the plate at a dilution of 1/100 in dilution buffer and incubated for 1 h, washed further five times with PBS, pH 7.4, and incubated for 1 h with goat anti-monkey (Rhesus macaque) HRP-labeled antibody (cat # a112767; Abcam), diluted 1:2,000 in PBS, pH 7.4. 100 $\mu$l per well of the OPD substrate solution (P9029-50G; Sigma-Aldrich) was added to the plate and incubated for 30 min away from light, and the reaction was stopped with 50 $\mu$l of sulfuric acid 3N. To detect the antigen–antibody reactivity, the plates were then read immediately at 492 nm in an ELx808 plate reader (BioTek).

#### *pLDH ELISA*

To measure *P. vivax* lactate dehydrogenase levels (PvLDH) in the monkey plasma samples, ELISA was performed using a matching pair of capture and detection antibodies (MyBioSource). Briefly, a 96-well microtiter plate was coated with mouse monoclonal anti-*Plasmodium* LDH (clone #M77288) at a concentration of 2 $\mu$g/ml in PBS (pH 7.4) and incubated overnight at 4°C. The plate was washed and incubated with blocking buffer (PBS-BSA 1%—reagent diluent) at RT for 2 h. After washing, samples were diluted 1:2, added to the plate, and incubated for 2 h. Next, plates were washed, and HRP-conjugated anti-pLDH detection antibody (clone #M12299), diluted 1:1,000 in blocking buffer, was incubated for 1 h at RT. Plates were washed and incubated for 15 min with substrate solution (OPD); the reaction was stopped by adding 2.5 M sulfuric acid. Optical density was determined at 450 nm. The cutoff of positivity was defined by correcting absorbance values generated in the plasma samples from blank values (plate controls). The total protein concentration from *P. vivax* schizont extracts was determined, and samples were used to perform standard curves ranging from 15,625 ng/ml up to 2,000 ng/ml. Lower absorbance values were in the range of O.D = 0.01–0.02. All positive monkey samples gave O.D. values equal to or higher than 0.05.

### Protein microarray and hybridization

All *P. vivax* sequences for the array used in this study were derived from the SAL-1 strain, which allowed the evaluation of greater breadth of antigens (but limited evaluation of antigenic variation). Antigens on this array were down-selected from larger arrays probed with reactive sera derived from various endemic regions. Only antigens demonstrating seroreactivity across all tested sera were included (104). The construction of the protein microarray was conducted using methods as described elsewhere (104). Briefly, coding sequences were PCR-amplified from *P. vivax* SAL-1 genomic DNA and cloned into the PXT7 plasmid using homologous recombinant as complete or overlapping fragments, the resulting plasmids (n = 244) were expressed in an *Escherichia coli*-based in vitro transcription/translation (IVTT) reactions, and the completed reactions were printed onto nitrocellulose-coated microarray slides (Grace Bio-Labs). Serum samples were diluted with 1/100 blocking buffer (Arrayit Corp) supplemented with the *E. coli* lysate (GenScript). The diluted serum samples were incubated with the protein arrays overnight at 4°C, followed by incubation with a goat anti-human IgG Texas Red secondary antibody (Southern Biotech). The arrays were scanned using a GenePix 4300A scanner (Molecular Devices) at 5-μm resolution and a wavelength of 594 nm (105).

### Protein microarray data processing and analysis

Raw median fluorescent intensity was local background–corrected using the normexp function (offset = 50, method = "mle," limma R package). All data were log-transformed (base 2) and normalized as a ratio of the signal for each spot to the mean of the no DNA control spot within each sample. The number of antigens that have reactivity above 0 in at least 10% of samples was calculated and included in the heatmap (generated in Microsoft Excel). Seropositive antigens for each sample were defined as those with reactivity above the mean of the sample-specific no DNA control spots + 3 SD. These seropositive antigens were totaled for each sample to determine the antigen breadth (number of reactive antigens). The antigen breadth AUC was calculated using the trapezoid rule after limiting the data to only the same number of time points for all inoculations. Pearson's correlations were performed for available ELISA titers and antigen breadth. All statistics and plots were done using R unless otherwise specified.

### PacBio whole-genome sequencing and analysis

*P. vivax* AMRU-1 and SAL-1 were amplified with selective whole-genome amplification (sWGA), using primers specific for the subtelomeres that are enriched for the low GC content (106). Amplified DNA was used for PacBio sequencing using a commercial protocol (GenScript). We obtained 373,772 subreads with an N50 (type of median length of the reads) of 13,119 bp for SAL-1 and 325,996 subreads with an N50 of 12,035 bp for AMRU-1. The reads were mapped with BWA-MEM for quality control (Fig 5A) and then assembled with canu (107) (parameter: genomeSize = 32 m; ErrorRate = 0.10; gnuplotTested = true; useGrid = 0 -pacbio-raw, version January 2018). The assemblies generated 113 and 103 contigs with an N50 of 50 and 41 k, and the largest contig be 195 and 140 kb for SAL-1 and AMRU-1, respectively. For annotation, the assemblies were loaded

into Companion (47), using PvP01 as a reference strain (June 2018; Augustus cutoff set to 0.4). The genome and its annotation can be found at http://cellatlas.mvls.gla.ac.uk/Assemblies/.

For the Gephi analysis, we extracted all the genes annotated as *pir* from the two Companion runs, merged them with the *pir* genes of PvP01, and performed an all-against-all BLASTp (-F F, *E*-value 1 × 10^(-6)). The results were parsed into the open source software Gephi to produce Fig 5C. For graphical representation, a force atlas algorithm was run, and then, the global identity cutoff was set to 32%, and the Fruchterman–Reingold algorithm was run.

## Gene expression microarray and gene expression analysis

### pir *gene probe development*

Sequences from sWGA, representing mostly the AT-rich subtelomeres and excluding the mitochondrial genes, were used as input for probe design using OligoRankPick (108) (oligo size = 60, % GC = 40). The oligos that were overlapping with core genome oligos from the existing *P. vivax* microarray (46) were removed (12 for SAL-1 and 6 for AMRU-1). The final list of new oligos contains 929 SAL-1 probes and 701 AMRU-1 probes, among which eight match two SAL-1 genes and five match two AMRU-1 genes (the full list is shown in Table S4).

### RNA preparation and microarray hybridization

Cell pellets from the blood samples collected at different time points during SAL-1 or AMRU-1 inoculations were stored in TRIzol. RNA was extracted and processed to be run in a customized microarray assay detecting both core and subtelomeric genes. The previously described microarray hybridization protocol was used for this study, with several modifications (109). In brief, 100 ng of cDNA was used for the subsequent 10 rounds of amplification to generate aminoallyl-coupled cDNA for the hybridizations as described previously (109). 17 μl (~5 μg) of each Cy5-labeled (GE Healthcare) cDNA of the sample and an equal amount of Cy3-labeled (GE Healthcare) cDNA of the reference pool were then hybridized together on a customized microarray chip using a commercially available hybridization platform (Agilent) for 20 h at 70°C with gentle rotation. Microarrays were washed and immediately scanned using Power Scanner (Tecan) at 10-μm resolution and with automated photomultiplier tubes gain adjustments to balance the signal intensities between both channels. The reference pool used for microarray was a mixture of 3D7 parasite strain RNA collected every 6 h during 48 h of the full IDC.

### Microarray analysis

To quantify microarray data signals, intensities were first corrected using an adaptive background correction using the method "normexp" and offset 50 using the Limma package in R (110). Next, we performed the within-array loess normalization followed by quantile normalization between samples/arrays. Each gene expression was estimated as the average of log₂ ratios (Cy5/Cy3) of representative probes; thus, intensities or log ratios could be comparable across arrays. Finally, probes with signal showing the median foreground intensity less than twofold of the median background intensity at either Cy5 (sample RNA) or Cy3 (reference pool RNA) channel were assigned missing values. Fold changes and

standard errors of relative gene expression were estimated by fitting a linear model for each gene, followed by empirical Bayes smoothing to the standard errors. Next, the average $\log_2$ expression level for each gene across all the arrays was calculated using the topTable function of the limma package. In parallel, we adjusted *P*-values for multiple testing using the Benjamini and Hochberg method to control the false discovery rate. The lists of differentially expressed genes for each of the comparisons were extracted by defining a cutoff of adjusted *P*-values < 0.001 and fold change > 1. The log fold change and adjusted *P*-values were graphed in volcano plots, using the EnhancedVolcano package in R (Table S5).

## Code Availability Statement

All R scripts for the microarray and correlation analysis and figure generation can be found at https://github.com/joaolsf/Vivax_immunizations_project.

## Supplementary Information

## Acknowledgements

Funding for this study was provided in part by core funds from the Gorgas Memorial Institute of Health Studies and the Sistema Nacional de Investigación de Panamá, SENACYT-SNI awarded to N Obaldía, Panamá, Panamá. This work was also supported by Royal Society Wolfson Research Merit Award (to M Marti) and Wellcome Trust Center Award 104111 (to F Achcar, JL Da Silva Filho, TD Otto, and M Marti).

### Author Contributions

N Obaldía: conceptualization, resources, data curation, formal analysis, supervision, funding acquisition, validation, investigation, visualization, methodology, project administration, and writing—original draft, review, and editing.
JL Da Silva Filho: data curation, formal analysis, investigation, visualization, and writing—review and editing.
M Núñez: investigation.
KA Glass: formal analysis and investigation.
T Oulton: formal analysis and investigation.
F Achcar: investigation.
G Wirjanata: investigation.
M Duraisingh: investigation.
P Felgner: resources and investigation.
KKA Tetteh: resources, data curation, formal analysis, investigation, and visualization.
Z Bozdech: resources and investigation.
TD Otto: resources, data curation, formal analysis, investigation, and visualization.
M Marti: conceptualization, resources, data curation, formal analysis, funding acquisition, investigation, visualization, methodology, and writing—original draft, review, and editing.

### Conflict of Interest Statement

The authors declare that they have no conflict of interest.

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
