## [Reviewer comments · Life Science Alliance]

Life Science Alliance

Sterile protection against vivax malaria by repeated blood stage infection in the Aotus monkey model

Nicanor Obaldia, Joao Da Silva Filho, Marlon Núñez, Katherine Glass, Tate Oulton, Fiona Achcar, Grennady Wirjanata, Manoj Duraisingh, Philip Felgner, Kevin Tetteh, Zbynek Bozdech, Thomas Otto, and Matthias Marti

DOI: <https://doi.org/10.26508/lsa.202302524>

Corresponding author(s): Matthias Marti, University of Glasgow and Nicanor Obaldia, Instituto Conmemorativo Gorgas

Review Timeline:

Submission Date:	2023-12-13
Editorial Decision:	2023-12-15
Revision Received:	2023-12-18
Accepted:	2023-12-19

Transaction Report:

Please note that the manuscript was reviewed at *Review Commons* and these reports were taken into account in the decision-making process at *Life Science Alliance*.

Review
COMMONS

1. General Statements [optional]

The two reviewers are very positive and emphasize the relevance of this study. Reviewer 1 notes that “the humoral immune responses but also parasite transcriptomics data is examined for the first time”. Reviewer 2 notes that our study “tries to mimic the infection in nature by reinfecting the Aotus monkeys with different stains of the parasite and then assesses the immune response with main emphasis on antibody response to the infection. This model is important to facilitate vaccine development and understanding the immune response against particular vaccines.”

Both reviewers only suggest text changes.

This section is mandatory. Please insert a point-by-point reply describing the revisions that were already carried out and included in the transferred manuscript.

Below please find the comments from reviewers 1 and 2 and our responses in italic, including text changes in the manuscript.

Reviewer #1 (Evidence, reproducibility and clarity (Required)):

To elucidate whether humoral immunity and/or genetic polymorphisms contribute to the protection against *P. vivax* blood-stage infection, the Authors assessed whether *P. vivax* strain-transcendent immunity can be achieved by repeated infection in *Aotus* monkeys. They infected six *Aotus* monkeys with blood stages of the *P. vivax* Salvador 1 (SAL-1) strain until obtaining sterile immunity, and then challenged with the heterologous AMRU-1 strain. Sterile immunity was achieved after two homologous infections, and partial protection against a heterologous AMRU-1 challenge was also achieved. IgG levels against parasite lysate by ELISA and protein microarray increased with repeated infections and correlated with the level of homologous protection.

Although there were transcriptional differences in the *P. vivax* gene repertoire between SAL-1 and AMRU-1, there was no evidence of major antigenic switching upon homologous or heterologous challenge. These findings suggest that the partial protection observed during heterologous challenge is caused by genetic polymorphism between strains, rather than immune evasion by antigenic switching.

Major Comments:

1) Title

Full Revision

There are several non-human primate models, therefore, please specify "Aotus monkey model" in the title.

Concur: We have added "Aotus monkey model" to the title.

2) Protein array

Lines 373-374 "against major immunogenic blood stage antigens (Ags)" Please add selection criteria for how they select these 244 antigens.

We have added the following paragraph in the methods to address this comment:

"All P. vivax sequences for the array used in this study were derived from the SAL-1 strain, which allowed the evaluation of greater breadth of antigens (but limited evaluation of antigenic variation). Antigens on this array were down selected from larger arrays probed with reactive sera derived from various endemic regions. Only antigens demonstrating seroreactivity across all tested sera were included [105]."

They prepared protein array (n=244) based on the SAL-1 sequence. Please add a discussion of how the data was affected by the sequence difference between SAL-1 and AMRU-1 strains. They described this point only on the top 7 targets (Lines 283-287). Any further difference in antibody reactivity between polymorphic and conserved antigens (SAL-1 and AMRU-1).

We agree with this concern and have added two sentences to the discussion.

"In comparing the SAL-1 and AMRU-1 strains to the PvP01 reference strain, the sequence data demonstrated clear differences between the isolates in the whole genome analysis. Therefore, this suggests that the current iteration of the microarray (n=244) used in the study did not capture the sequence target(s) responsible for the partial protection observed."

Please also add a discussion on how they can interpret their protein microarray data because the E. coli-based IVTT proteins array detects antibody responses against linear epitopes of the printed antigens.

The IVTT cell-free E.coli express system used to generate the protein microarrays represents an unbiased systems biology approach to antigen identification (Davies DH et al PMID: 26428458). The focus is intentionally on linear epitopes as attempting to capture correctly folded whole proteins is a notoriously difficult venture (Vedadi M et al Mol Biochem Parasitol. PMID: 17125854; Mehlin C et al. Mol Biochem Parasitol. PMID: 16644028). The system has shown proven utility across several disease in identifying important antigenic targets which can then be explored in greater detail using other methods (Wager LE et al. Nat Med. PMID: 33432170;

Full Revision

Nakajima R et al. *mSphere* PMID: 30541779; Virgil A et al. *Future Microbiol.* PMID: 20143947; Vankatesh A et al *Sci Rep.* PMID: 35654904).

The following text and references have been included into the discussion:

*“This approach was supported by previous studies which demonstrated the utility of the IVTT platform in high throughput antigen discovery across several disease areas (Jan S et al. *Front Immunol* PMID: 37533862; Nakajima R et al. *mSphere* PMID: 30541779; King CL et al. *Am J Trop Med Hyg* PMID: 26259938; Vankatesh A et al. *Methods Mol Biol.* PMID: 34115357; Vankatesh A et al. *Malar J* PMID: 30995911).”*

3) Weakness

Please summarize the weak points of this study (i.e. small number of animals used) in the Discussion section.

We have added and combined a few phrases with limitations in the discussion section:

“The partial protection observed in the heterologous AMRU-1 challenges may therefore be due to major genetic differences and hence antibody epitope variation between the two strains [50]. In comparing the SAL-1 and AMRU-1 strains to the PvP01 reference strain, the sequence data demonstrated clear differences between the isolates in the whole genome analysis. Therefore, this suggests that the current iteration of the microarray (n=244) used in the study did not capture the sequence target(s) responsible for the partial protection observed. To overcome this limitation and induce high levels of protective antibodies, we propose use of an immunization regime with whole parasite antigen pools from a mixture of genetically diverse strains. Another limitation of this study is the small number of subjects. The study can be considered as exploratory (i.e. looking for patterns of response rather than hypothesis testing [95]), hence the number of subjects used in the only group studied is typical of such exploratory research with humans [35, 96] and NHP [38].”

Minor Comments:

4) Line 129 "inoculation level II" Please reword this to "2nd inoculation" throughout the manuscript because "inoculation level" is a bit confusing for the readers.

Do not concur: It is easier to understand, in the figures in particular. Unless the editor insists, we would rather keep as is.

5) Line 320 "pir genes" Please spell out because this is the first appearance in this manuscript.

Done. Plasmodium interspersed repeat (PIR) genes.

6) Line 373 "IVTT" Please spell out because this is the first appearance in this manuscript.

Full Revision

Done. in vitro transcription/translation reaction (IVTT)

7) Line 404 "VIR antigens" Please spell out because this is the first appearance in this manuscript.

Done. Plasmodium vivax interspersed repeat (VIR) antigens.

8) Line 498 "Goat anti-monkey Rhesus macaque)" This may be HRP-labelled? Please correct.

Concur: We have added HRP labelled to: "Goat anti-monkey Rhesus macaque HRP-labelled"

9) Line 512 "temperature Plates" should be "temperature. Plates".

10) Line 514 "sulphuric acid 2.5M" should be "2.5M sulphuric acid".

Concur: Changed to "2.5M sulphuric acid".

11) Line 516 "Plasmodium falciparum" should be "Plasmodium vivax".

Concur: Changed to "Plasmodium vivax".

12) Line 524 "Escherichia.coli" should be "Escherichia coli".

Concur: Changed to "Escherichia coli".

13) Line 604 "is spleen-dependent (ref)" Please add a reference.

This paragraph has been removed as the data are not included in this study.

14) Line 1099 "core genes" Please add a description of what core genes mean.

Has now been added in the text line 319.

15) Figure S2 Please label each panel in Figure S2 A&B. Maybe I, II, III, IV from the left. Please also revise the label of the X-axis in Figure S2C because "Inoculation level" is misleading.

We have added the labeling to S2A and B.

****Referees cross-commenting****

I agree with Reviewer#2 comments.

Reviewer #1 (Significance (Required)):

1) General assessment:

This is a valuable and important study conducted by qualified experts in this research field. All the works were carefully designed, and clearly presented, and the manuscript is well written.

(1) Strongest and most important aspects?

Aotus monkey study with intensive data acquisition including humoral immune response and detailed parasite transcriptomic investigation.

(2) Weakness

The number of animals used is rather small.

2) Advance:

Does the study extend the knowledge in the field and in which way?

Not only the humoral immune responses but also parasite transcriptomics data is examined for the first time.

3) Audience:

Malariologists will be interested in or influenced by this research

The data in this study will be the basis of future whole-parasite-based vaccine development.

My field of expertise is malariology and malaria vaccine research.

Reviewer #2 (Evidence, reproducibility and clarity (Required)):

This study focuses on the development of the model which can be further used as the model for developing a vaccine for the malaria parasite *Plasmodium vivax*. The researchers infected Aotus monkeys with one strain, achieved immunity, and then exposed them to a different strain. Four monkeys became immune to the initial strain, and three showed partial protection against the second strain. The researchers found that differences in genetic diversity and gene expression between strains are responsible for the varying levels of protection. This study provides insights for testing candidate vaccines against *P. vivax*. This model is unique and important for facilitating vaccine developments.

- The researchers provide a clear methodology and suitable for the proposed research questions.

- Did researchers observed any gametocytes after inoculations especially in the asymptomatic one or the prolong parasitemia. If they found, whether those gametocyte are infectious?

We did not focus on gametocytes in this study, hence no mosquito infection experiments were performed.

Full Revision

Reviewer #2 (Significance (Required)):

The asymptomatic infections are common in malaria endemic areas but it is hard to identify the underlying immune mechanism in response to the disease. This model tries to mimic the infection in nature by re-infecting the Aotus monkeys with different strains of the parasite and then assesses the immune response with main emphasis on antibody response to the infection. This model is important to facilitate vaccine development and understanding the immune response against particular vaccines.

December 15, 2023

RE: Life Science Alliance Manuscript #LSA-2023-02524-T

Matthias Marti
University of Glasgow
Wellcome Centre for Molecular Parasitology
Glasgow
United Kingdom

Dear Dr. Marti,

Thank you for submitting your revised manuscript entitled "Sterile protection against *Plasmodium vivax* malaria by repeated blood stage infection in the Aotus monkey model". We would be happy to publish your paper in Life Science Alliance pending final revisions necessary to meet our formatting guidelines.

- please add a Running Title and a Summary Blurb/Alternate Abstract to our system
- please add ORCID ID to the secondary corresponding author--they should have received instructions on how to do so
- please add a Category for your manuscript in our system
- please add the Twitter handle of your host institute/organization as well as your own or/and one of the authors in our system
- please add Author Contributions to our system as well
- please add your main, supplementary figure, and table legends to the main manuscript text after the references section
- please remove figures from the manuscript text; they should be uploaded separately, and their legends should appear only in the manuscript text
- please add callouts for Figures 2C, S2C and table S4 to your main manuscript text
- please remove the Author Summary section, and incorporate those comments into other sections, if needed

A. FINAL FILES:

B. MANUSCRIPT ORGANIZATION AND FORMATTING:

Sincerely,

Reviewer #1 (Comments to the Authors (Required)):

All the comments from 2 Reviewers have been appropriately addressed, and the revised manuscript is sufficiently developed.

December 19, 2023

RE: Life Science Alliance Manuscript #LSA-2023-02524-TR

Prof. Matthias Marti
University of Glasgow
Wellcome Centre for Integrative Parasitology
Glasgow
United Kingdom

Dear Dr. Marti,

Thank you for submitting your Research Article entitled "Sterile protection against vivax malaria by repeated blood stage infection in the Aotus monkey model". It is a pleasure to let you know that your manuscript is now accepted for publication in Life Science Alliance. Congratulations on this interesting work.

DISTRIBUTION OF MATERIALS:

Again, congratulations on a very nice paper. I hope you found the review process to be constructive and are pleased with how the manuscript was handled editorially. We look forward to future exciting submissions from your lab.

Sincerely,
